# Looking at New Unexpected Disease Targets in *LMNA*-Linked Lipodystrophies in the Light of Complex Cardiovascular Phenotypes: Implications for Clinical Practice

**DOI:** 10.3390/cells9030765

**Published:** 2020-03-20

**Authors:** Héléna Mosbah, Camille Vatier, Franck Boccara, Isabelle Jéru, Olivier Lascols, Marie-Christine Vantyghem, Bruno Fève, Bruno Donadille, Elisabeth Sarrazin, Sophie Benabbou, Jocelyn Inamo, Stéphane Ederhy, Ariel Cohen, Barbara Neraud, Pascale Richard, Fabien Picard, Sophie Christin-Maitre, Alban Redheuil, Karim Wahbi, Corinne Vigouroux

**Affiliations:** 1Inserm UMRS938, Saint-Antoine Research Center, Sorbonne University, 75012 Paris, Francebruno.feve@aphp.fr (B.F.);; 2Reference Center of Rare Diseases of Insulin Secretion and Insulin Sensitivity (PRISIS), Department of Endocrinology, AP-HP Saint-Antoine Hospital, 75012 Paris, France; 3National Institute of Health and Medical Research, Department of Cardiology, AP-HP Saint-Antoine Hospital, 75012 Paris, France; 4Department of Molecular Biology and Genetics, AP-HP Saint-Antoine Hospital, 75012 Paris, France; 5Inserm U1190—EGID (European Genomic Institute for Diabetes), Endocrinology, Diabetology and Metabolism Department, CHU Lille, 59000 Lille, France; 6Reference Center of Neuromuscular Rare Diseases, CHU Fort de France, Pierre Zobda Quitman Hospital, 97200 Martinique, France; 7Cardiology Department, CHU Fort de France, Pierre Zobda Quitman Hospital, 97200 Martinique, Francejocelyn.inamo@chu-martinique.fr (J.I.); 8Inserm UMRS1166, Sorbonne University, 75013 Paris, France; 9Department of Internal Medecine, Foch Hospital, 92150 Suresnes, France; 10AP-HP Pitié Salpêtrière-Charles Foix University Hospital, Functional Unit of Molecular and Cellular Cardiogenetics and Myogenetics, Department of Metabolic Biochemistry, 75013 Paris, France; 11AP-HP Cochin Hospital, Cardiology Department, FILNEMUS, Paris-Descartes, Sorbonne Paris Citeé University, 75006 Paris, France; 12AP-HP/LIB (INSERM-CNRS- Sorbonne University)/ICAN Imaging Core Lab, Institute of Cardiology, AP-HP Pitié Salpêtrière Hospital, Department of Cardiovascular Imaging, 75013 Paris, France; 13Paris Cardiovascular Research Centre (PARCC), Inserm UMRS970, 75015 Paris, France

**Keywords:** lipodystrophy, *LMNA*, cardiovascular disease

## Abstract

Variants in *LMNA*, encoding A-type lamins, are responsible for laminopathies including muscular dystrophies, lipodystrophies, and progeroid syndromes. Cardiovascular laminopathic involvement is classically described as cardiomyopathy in striated muscle laminopathies, and arterial wall dysfunction and/or valvulopathy in lipodystrophic and/or progeroid laminopathies. We report unexpected cardiovascular phenotypes in patients with *LMNA*-associated lipodystrophies, illustrating the complex multitissular pathophysiology of the disease and the need for specific cardiovascular investigations in affected patients. A 33-year-old woman was diagnosed with generalized lipodystrophy and atypical progeroid syndrome due to the newly identified heterozygous *LMNA* p.(Asp136Val) variant. Her complex cardiovascular phenotype was associated with atherosclerosis, aortic valvular disease and left ventricular hypertrophy with rhythm and conduction defects. A 29-year-old woman presented with a partial lipodystrophy syndrome and a severe coronary atherosclerosis which required a triple coronary artery bypass grafting. She carried the novel heterozygous p.(Arg60Pro) *LMNA* variant inherited from her mother, affected with partial lipodystrophy and dilated cardiomyopathy. Different lipodystrophy-associated *LMNA* pathogenic variants could target cardiac vasculature and/or muscle, leading to complex overlapping phenotypes. Unifying pathophysiological hypotheses should be explored in several cell models including adipocytes, cardiomyocytes and vascular cells. Patients with *LMNA*-associated lipodystrophy should be systematically investigated with 24-h ECG monitoring, echocardiography and non-invasive coronary function testing.

## 1. Introduction

Pathogenic variants in the *LMNA* gene are responsible for several rare clinical disorders collectively called laminopathies. Laminopathies affect either striated muscle (skeletal and cardiac muscle dystrophy), adipose tissue (lipodystrophies), peripheral nerve (Charcot Marie Tooth axonal neuropathy) or multiple organs with clinical signs of accelerated ageing (progeroid syndromes) [1]. The clinical heterogeneity of laminopathies mirrors the multiple roles of the *LMNA*-encoded A-type lamins, which are ubiquitously expressed essential components of the cell nucleus. Together with B-type lamins, A-type lamins form the lamina meshwork at the nucleoplasmic side of the inner nuclear membrane and are also found as nucleoplasmic intermediate filaments [2]. A-type lamins are able to interact with nuclear lipids, with a number of proteins, and with chromatin, and display a number of structural and functional roles, contributing to nuclear and cytoskeletal organization, cell mechanical properties, mechanotransduction pathways, chromatin organization, regulation of gene expression, genome stability and cell differentiation [2].

Although laminopathies are mainly tissue-specific diseases, the heart and/or vascular wall appear as recurrent phenotypic targets. Indeed, laminopathies of the striated muscle, including Emery–Dreifuss muscular dystrophy, isolated dilated cardiomyopathy and limb-girdle muscular dystrophy 1B, actually represent a spectrum of overlapping phenotypes, with dilated cardiomyopathy as the unifying feature [3]. Dunnigan-type familial partial lipodystrophy (FPLD2) is characterized by partial lipoatrophy with insulin resistance, hypertriglyceridemia, liver steatosis, but also by frequent and early atherosclerotic events [4,5,6]. In Hutchinson–Gilford progeria syndrome (HGPS) and *LMNA*-associated progeroid syndromes, multitissular features of accelerated ageing, including lipoatrophy, are associated with major atherosclerosis and/or valvular calcifications [7,8,9]. Cardiovascular involvement, according to this classification of laminopathies, is mainly expressed as cardiomyopathy with rhythm and conduction disturbances in laminopathies of the striated muscle, and as arterial wall dysfunction and/or valvulopathy in lipodystrophy-associated laminopathies such as FPLD2 and progeroid syndromes.

This dichotomy in cardiovascular phenotypes was suggested by a hierarchical cluster analysis of genotype–phenotype correlations of *LMNA*-associated diseases [10]. *LMNA* pathogenic variants causing striated muscle diseases are distributed throughout all of the exons of the gene (http://www.umd.be/LMNA). In contrast, the typical FPLD2 phenotype is mainly due pathogenic variants in exon 8 of the gene, leading to amino acid substitutions that alter the surface charge of the immunoglobulin-like fold of the A-type lamins tail domain. Progerin, the mutant form of prelamin A responsible for HGPS, is a truncated protein with a farnesylated C-terminal cysteine resulting from the expression of a specific *LMNA* variant in exon 11. The tissue-specificity of the different laminopathies could thus be based on distinct pathophysiological mechanisms [11]. Historically, it was proposed that *LMNA* pathogenic variants induce either mechanical defects of the lamina, which especially affect tissues submitted to intense mechanical stress, such as skeletal and cardiac muscles (“structural model” of laminopathies), or disruption of chromatin organization and function, which cause tissue-specific deregulation of gene expression (“gene expression model”) [2]. In addition, a third pathophysiological model of laminopathies states that progerin displays multiple toxic cellular effects leading to premature cell senescence, particularly in vascular cells [12].

The specificity of laminopathic cardiovascular alterations usually translates into standardized clinical approaches, with investigations directed either toward heart function and electrical activity, or toward cardiac and peripheral large vessels, depending on the laminopathy global phenotype.

However, rare reports have pointed out more complex clinical situations. Cardiomyopathic features have been reported in a few patients with typical FPLD2 carrying the most frequent p.(Arg482Trp) or p.(Arg482Gln) *LMNA* pathogenic variants affecting the immunoglobulin-like C-terminal domain of A-type lamins [13]. Most forms of mixed laminopathy phenotypes with lipodystrophy and cardiomyopathy have been described in patients with non p.Arg482 *LMNA* pathogenic variants affecting different protein domains [13,14,15].

We further illustrate the complexity of cardiovascular laminopathic phenotypes by two clinical cases. In these patients referred for lipodystrophy, extensive investigations revealed multiple, potentially life-threatening cardiovascular laminopathic impairments, leading to specific therapeutic approaches. Besides the need of cardiovascular clinical care in patients with lipodystrophic laminopathies, these observations suggest that further studies should search for other potential overlapping symptoms between metabolic and striated muscle laminopathies. In addition, the pathophysiological mechanisms by which some *LMNA* variants could affect adipocytes, cardiomyocytes and/or vascular cells should be investigated.

## 2. Patients and Methods

We report the clinical cases of two patients referred to our National Reference Center of Rare Diseases of Insulin Secretion and Insulin Sensitivity (PRISIS), Paris, France.

The study has been approved by the local ethical committee and has been performed according to local and European Union ethical rules.


**Case 1**


A 33-year-old West Indian woman was referred to our department for lipodystrophy and diabetes with severe insulin resistance (treated with 6 IU/kg of insulin/day).

Her mother suddenly died from an unspecified cardiac cause at age 48 (subject I-2, Figure 1). Her older stepbrother, known to harbor a *LMNA* c.407A>T p.(Asp136Val) variant in exon 2 affecting the central rod domain of A-type lamins, died from cardiac failure at age 44 (subject II-1, Figure 1). Both were described with generalized lipoatrophy. Her younger brother, also known to harbor the *LMNA* p.(Asp136Val) variant, was described to have heart conduction abnormality and generalized lipoatrophy (subject II-5, Figure 1). Her older sister, described with generalized lipoatrophy, refused any genetic investigation (subject II-2, Figure 1). This latter patient had a 16-year old son, known to carry the same *LMNA* variant, who recently died from heart failure (subject III-1, Figure 1).

At the age of 28, the proband presented with an acute pancreatitis due to severe hypertriglyceridemia (triglycerides peak: 6.8 mmol/L, N < 1.7) and subsequently suffered from frequent gastrointestinal symptoms including vomiting and diarrhea.

She displayed several facial dysmorphic features, i.e., beaked nose, micrognathia and dental crowding, prominent eyes, loss of frontal hair, and arachnodactyly. Skin appeared as sclerodermatous with mottled depigmentation areas in abdomen and limbs. Lipoatrophy was severe and generalized with extremely low body mass index (BMI) (11.7 kg/m^2^ with weight of 30 kg and height of 160 cm) and body fat mass (1200 g, i.e., 4% of total body weight) as assessed by Dual-Energy X-Ray Absorptiometry (DEXA). Serum leptin and adiponectin were strongly decreased (0.8 ng/mL and 0.8 mg/L, respectively), while glycosylated hemoglobin and serum triglycerides were increased (7.7% and 12 mmol/L, respectively). Abdominal echography was in favor of liver steatosis although liver enzyme levels were normal.

She suffered from dyspnea and had a systolic aortic heart murmur. She was a non-smoker. Her blood pressure was normal. The resting electrocardiogram (ECG) showed an incomplete left bundle branch block, left and right atrial hypertrophy, and left ventricular (LV) hypertrophy (Figure 2a). Cardiac ultrasound revealed a moderate aortic valve stenosis with mild regurgitation (aortic valve area: 0.82 cm^2^, i.e., 0.69 cm^2^/m^2^ of body surface, maximum aortic velocity: 2.7 m/s, gradient of aortic transvalvular pressure: 15 mmHg). The coronary computed tomography (CT) angiogram revealed extensive calcifications of ascending aorta and aortic valve, without significant coronary calcification or stenosis (Agatson score: 2). The abdominal CT angiography found major atherosclerosis and calcifications of abdominal vessels. Cardiac magnetic resonance imaging (MRI) showed a LV concentric remodeling (LV concentricity = 1.2 g/mL) with hypertrophy (septal thickness = 11 mm and LV mass index = 101 g/m^2^) (Figure 2b), and a LV systolic dysfunction (LV ejection fraction or EF: 42%). A diffuse hypokinesis was observed along with antero-septo-apical akinesia and wall thinning (Figure 2c) with heterogeneous late gadolinium enhancement, compatible with myocardial fibrosis [16] (Figure 2d,e and Appendix A). The systematic 24-h ECG monitoring revealed malignant sustained ventricular tachycardia. The patient declined the implantation of a cardioverter-defibrillator.

Two years later, she became severely dyspneic with major aortic valve stenosis (aortic valve area: 0.55 cm^2^ i.e., 0.44 cm^2^/m^2^ of body surface, maximum aortic velocity: 3.5 m/s, gradient of aortic transvalvular pressure: 43 mmHg). A transcatheter aortic valve implantation (TAVI) was successfully performed.

Her genetic analysis revealed a c.407A >T *LMNA* heterozygous missense variant, predicting the p.(Asp136Val) aminoacid substitution previously observed in her brother.


**Case 2**


A 29-year-old Caucasian woman was referred to our department for partial lipodystrophy.

Her mother was previously diagnosed with dilated cardiomyopathy with rhythm and conduction disturbances at age 44, and with partial lipodystrophy, diabetes, hypertriglyceridemia and liver steatosis at age 45. She had been implanted with a cardioverter-defibrillator at age 46 and underwent a heart transplant at age 48. She was not known to display any coronary artery disease. Her genetic analysis revealed a heterozygous *LMNA* c.179G > C p.(Arg60Pro) variant in exon 1 affecting the protein central rod domain (coil 1A).

The proband had suffered from gestational diabetes (treated with diet only) at age 21. She presented with partial lipodystrophy which became clinically obvious in the pubertal period with excess accumulation of fat in the face and neck, loss of fat from the arms, legs and trunk, generalized muscular hypertrophy and axillary *acanthosis nigricans*. At age 24, her fasting glycemia was normal, but insulin resistant diabetes was diagnosed based on an oral glucose tolerance test (glucose and insulin were respectively 5.5 mmol/L and 49 μIU/L at fast, 14.3 mmol/L and 552 μIU/L at 120min). At that time, she had been diagnosed with the same *LMNA* heterozygous variant as her mother (Figure 3a,b).

At the time of examination, her BMI was 24 kg/m^2^ (weight, 62 kg and height, 160 cm). Her fat mass percentage was moderately decreased (18% fat mass using DEXA) concordant with the serum leptin concentration (6.8 ng/mL). She also had fatty liver disease (as assessed by echography) with increased levels of alanine transaminase (ALT 82 IU/L, N < 32) and gamma glutamyl transpeptidase (GGT 85 IU/L, N < 32). She had dyslipidemia with increased serum triglycerides (3.4 mmol/L), decreased HDL-cholesterol (0.8 mmol/L) and normal LDL-cholesterol (2.6 mmol/L) under statin therapy. Her glycosylated hemoglobin was under 7%, indicating an adequate glycemic control under diet and metformin only. She did not present any microvascular complication of diabetes.

She was a non-smoker and her blood pressure was normal. Although she did not complain of any functional symptoms, cardiac investigations first aimed to detect a potential cardiomyopathy. The resting ECG was normal. The 24-h Holter monitoring showed atrial premature complexes without atrial fibrillation; PR, QRS and QT intervals were normal. The electrophysiological study did not show any conduction disturbance. The left ventricular ejection fraction was normal (LV ejection fraction: 65% assessed by echocardiography) and the cardiac MRI did not show any sign of dilated cardiomyopathy. A systematic ECG stress test was performed in this context of diabetes and lipodystrophy. The ECG stress test was stopped, as the patient developed breathlessness with a peak heart rate of 155 bpm (81% of the theorical maximum heart frequency), at 5.8 METs of work. In addition, down-sloping ST-segment depressions were observed in II, III, aVF and V3-V6 leads. The coronary angiogram revealed severe proximal and diffuse stenosis of the coronary artery territories, with ostial stenosis of both the right coronary artery (Figure 4A) and the left main coronary artery (LMCA) (Figure 4B–D). Intravascular ultrasound found a LMCA significant stenosis with circumferential calcifications (Figure 4a), and a severely decreased lumen area (4.73 mm^2^) (Figure 4b). The proximal left anterior descending artery was calcified (Figure 4c,d) whereas the mid part showed minimal atheroma (Figure 4e). The patient underwent a successful triple coronary artery bypass graft surgery.

## 3. Discussion

These two clinical cases represent two striking examples of overlapping cardiovascular phenotypes associated with *LMNA*-linked lipodystrophies due to non p.Arg482 pathogenic variants. The first patient had a family history of sudden cardiac death, a situation that should catch attention and lead to careful cardiac investigation as previously emphasized [13]. In the second patient, an early severe coronary disease occurred despite no striking cardiometabolic risk factor.

The first patient, diagnosed with a progeroid lipoatrophy, illustrates the cardiovascular phenotype complexity of laminopathies. Indeed, she presented with atherosclerosis, aortic valvular disease and mild left ventricular hypertrophy with rhythm and conduction system defects due to a novel heterozygous *LMNA* p.(Asp136Val) pathogenic variant. We only found one study reporting a pathogenic variant at the same codon p.(Asp136His). In this latter report, the patient had a generalized lipodystrophy with progeroid features, diabetes, acute pancreatitis and liver steatosis, no valvular disease but left ventricular hypertrophy without any other information about the cardiac phenotype. Other *LMNA* pathogenic heterozygous missense variants in the vicinity of Asp136 site, most of them modifying the charge of an aminoacid (*i.e.* (p.(Arg133Leu), p.(Glu138Lys), p.(Leu140Arg), p.(Glu145Lys)) have been shown to be responsible for progeroid syndromes; major atherosclerosis, valvular calcifications and/or left ventricular dysfunction have been described in some patients, but rhythm and conduction defects were not reported [17].

Premature atherosclerosis is a striking feature of HGPS, which strongly contributes to the early mortality of patients with this typical, very severe form of progeria [9]. The autopsy of cardiovascular tissues from patients with HGPS reveals features of aging-related atherosclerosis, arteriolosclerosis of small vessels and prominent adventitial fibrosis [18]. HGPS is due to the synthesis of progerin, which arises from abnormal *LMNA* splicing and subsequent abnormal post-translational maturation of mutated prelamin A. Progerin remains permanently farnesylated and forms an abnormal lamina network, tightly bound to the inner nuclear membrane [19,20]. Therapeutic trials using farnesyl transferase inhibitors showed evidence of cardiovascular benefit in patients with HGPS, in favor of a prominent pathophysiological role of the abnormal farnesyl moiety of progerin present in vascular cells [17,21], which is also supported by murine models [12,22,23]. Several studies, including ours, have shown that other *LMNA* pathogenic variants, responsible for atypical progeroid syndrome or FPLD2, could also lead to abnormal prelamin A accumulation, which could contribute to vascular senescence and atherosclerosis by targeting endothelial and/or vascular smooth muscle cells [4,24,25,26,27]. However, other pathophysiological mechanisms are probably also involved since accumulation of farnesylated prelamin A is not observed in other cases of *LMNA*-associated lipodystrophy and/or progeroid syndromes [28,29].

Electrocardiographic abnormalities with cardiac repolarization abnormalities have been described in HGPS, which could result from a defective cardiomyocyte connectivity due to the mislocalization of the gap junction protein Connexin 43 [30]. Alterations in lamin-mediated regulation of AKT (protein kinase B)/mTOR (mammalian target of rapamycin) pathways, which influence the expression or localization of Connexin 43, could represent another potential mechanism for arrhythmogenicity [31]. To note, prelamin A accumulation has been shown to mediate myocardial inflammation, which could contribute to acquired or genetically-determined cardiomyopathies [32]. Finally, alterations in extracellular matrix leading to fibrosis were described in laminopathies affecting adipose tissue [33,34], the vascular wall [18] and/or heart muscle [35], and could participate in multitissular complications. In that setting, the myocardial late gadolinium enhancement observed on the cardiac MRI in Patient 1 suggests evidence of myocardial fibrosis secondary to microvascular ischemia and/or primary myocardial involvement, as previously described in *LMNA*-related cardiomyopathies [16]. In this cardiac MRI study, late gadolinium enhancement in favor of myocardial fibrosis was observed in 15 out of 17 patients with *LMNA*-related dilated cardiomyopathies. To note, the patients did not presented with progeroid features but harbored *LMNA* variants affecting the coil 1B of the central rod domain of type A lamins, as in Patient 1 [16].

Lipodystrophy-associated progeroid syndromes due to different *LMNA* pathogenic variants have been described with cardiomyopathy which could be worsened by a concomitant diabetes, as noted in Patient 1 [15]. It is striking to note that coronary arteries were not significantly stenotic in this patient. Therefore, we assume that ventricular hypertrophy could be due to the conjunction of microvascular ischemic lesions worsened by diabetes, aortic valve stenosis, and primitive laminopathic muscle disease in Patient 1.

Ventricular arrhythmia, as observed in Patient 1, is a major cause of death in cardiomyopathic laminopathies and may occur early, even without ventricular dysfunction [31,36]. A study pinpointed that only one half of the patients carrying pathogenic or likely pathogenic *LMNA* mutations had a left ventricular fractional shortening <50%, suggesting that *LMNA* cardiomyopathy often manifests as a primary arrhythmia independent of muscle disease [37]. The high risk of sudden death owing to rhythm and/or electrical conduction disturbances requires specific criteria for cardioverter-defibrillator implantation in laminopathies [38].

*LMNA*-associated lipodystrophies without progeroid features can also, although rarely, be associated with dilated [14,39,40,41,42] or hypertrophic cardiomyopathy [43,44]. To note, cardiac MRI in patients with congenital generalized lipodystrophy due to *BSCL1* or *BSCL2* pathogenic variants also show a concentric LV hypertrophy, independent of blood pressure, which could be due to increased myocardial triglyceride content [45]. In addition, epicardial fat has been shown to be increased in *LMNA*-associated partial lipodystrophy [46]. Therefore, whatever their underlying molecular cause, ectopic lipid storage and lipotoxicity-driven defects could contribute to cardiac hypertrophy in lipodystrophic syndromes.

In Patient 2, we did not expect to observe a severe coronary artery disease, given the phenotype of cardiac muscle laminopathy in her mother associated with a *LMNA* p.(Arg60Pro) pathogenic variant, without progeroid features. Mixed phenotypes of lipodystrophy and dilated cardiomyopathy have been described in patients with other *LMNA* variants affecting the N-terminal part of lamin A/C (p.(Arg28Trp), p.(Arg62Gly), and p.(Asp192Val)) [42,47]. In addition, several patients carrying the p.(Arg60Gly) *LMNA* variant at the same codon as our patient were reported. Ten affected patients from the same family were described with dilated cardiomyopathy and/or cardiac conduction defects. Of note, one of them developed a severe coronary artery disease in the transplanted heart [48]. In the study of Fatkin and colleagues, 4 out of 8 patients carrying the *LMNA* p.(Arg60Gly) variant presented with a dilated cardiomyopathy and 6 out of 8 had cardiac conduction defects requiring a pacemaker. Again, one of these patients died from coronary artery disease in the transplanted heart [49]. In Patient 2, it is unlikely that the severe coronary artery disease was a complication of diabetes, given her young age, the short duration of permanent diabetes and the adequate glycemic control. Furthermore, she did not present any other traditional cardiovascular risk factors. It is thus possible that the *LMNA* p.(Arg60Pro) variant could induce direct alterations of the vascular wall. In accordance, we have shown, using patient induced pluripotent stem cells, that the FPLD2-causing *LMNA* p.Arg482Trp variant elicits early mesodermal gene expression defects with altered endothelial differentiation [50], in favor of a developmental origin of both lipodystrophy and premature atherosclerosis. This hypothesis is strengthened by other experimental results showing that several disease-causing *LMNA* variants reconfigure chromatin spatial conformation and epigenetic patterns [51,52,53].

Nevertheless, in our patient, we cannot exclude that insulin resistant diabetes could also contribute to endothelial dysfunction, leading to atherosclerosis, as previously suggested in patients with FPLD2 due to *LMNA* p.Arg482 “hotspot” variants [5]. However, the inverse relationship could also be discussed since it was recently shown that endothelial dysfunction, through secreted factors, may directly alter adipose tissue homeostasis and lead to insulin resistance [54]. Direct evidence that endothelial cell senescence induces adipose tissue metabolic dysfunction and systemic insulin resistance was recently demonstrated in Tie2-TERF2DN-Tg mice, which display an endothelium-specific form of progeria [55].

In a series of 19 patients with typical FPLD2, clinical atherosclerosis was reported in 68% of them, and atherosclerosis was present before age 45 in 62%**,** although diabetes and low-density lipoprotein cholesterol were well-controlled [4]. Kwapich and colleagues collected cardiometabolic data of 58 patients with *LMNA* pathogenic variants. A total of 4 out of 20 patients screened for myocardial ischemia had significant coronary stenosis, and all of them were affected with the typical *LMNA* p.Arg482 variant [13]. In light of our observations, not only patients with the typical FPLD2 phenotype, prone to atherosclerosis as shown previously, but also patients with non-classical forms of laminopathic lipodystrophy should be specifically investigated to allow early diagnosis and cardiovascular management.

At the pathophysiological level, recent data suggest that the previously described “structural model” and “gene expression model” of laminopathies could interact, leading to an integrative unified disease model in which altered lamin-mediated mechanotransduction and chromatin regulation pathways are interrelated [11,56]. This concept should be further analyzed in order to decipher the mechanisms by which adipocytes, cardiomyocytes and/or vascular cells could be altered upon the expression of specific pathogenic *LMNA* variants.

## 4. Conclusions

These two clinical cases show that both cardiac vessels and cardiac muscle can be affected in *LMNA*-associated lipodystrophies. Some authors proposed a stratification of the risk of cardiomyopathy according to the type of *LMNA* mutations [57]. Nevertheless, clinicians must be very careful since overlapping cardiovascular phenotypes are associated with different pathogenic variants. The lipodystrophy syndromes practice guidelines remind us that in patients with atypical progeroid syndromes and FPLD2 due to *LMNA* mutations, cardiac abnormalities including ischemic heart disease, cardiomyopathy, arrhythmias and sudden death are reported [58]. We propose that patients with laminopathic lipodystrophy, whatever their underlying *LMNA* pathogenic variant, should be systematically investigated with 24-h ECG monitoring, echocardiography and/or cardiac MRI for the diagnosis of cardiomyopathy, and stress test and/or coronary CT angiogram to detect coronary artery disease.

## Figures and Tables

**Figure 1 cells-09-00765-f001:**
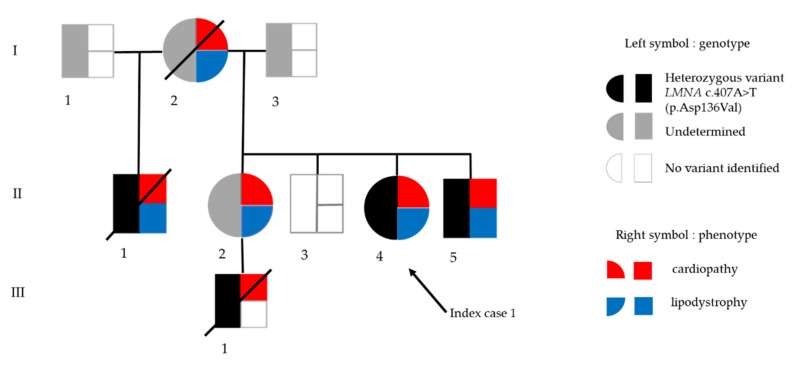
Genealogic tree of family in Patient 1. Black arrow indicates the index case. In the symbol for each subject, genotype is depicted on the left (black: presence of the heterozygous *LMNA* p.(Asp136Val) variant, white: absence of the *LMNA* variant, grey: genotype not determined), and phenotype is depicted on the right (red: cardiopathy, blue: lipodystrophy).

**Figure 2 cells-09-00765-f002:**
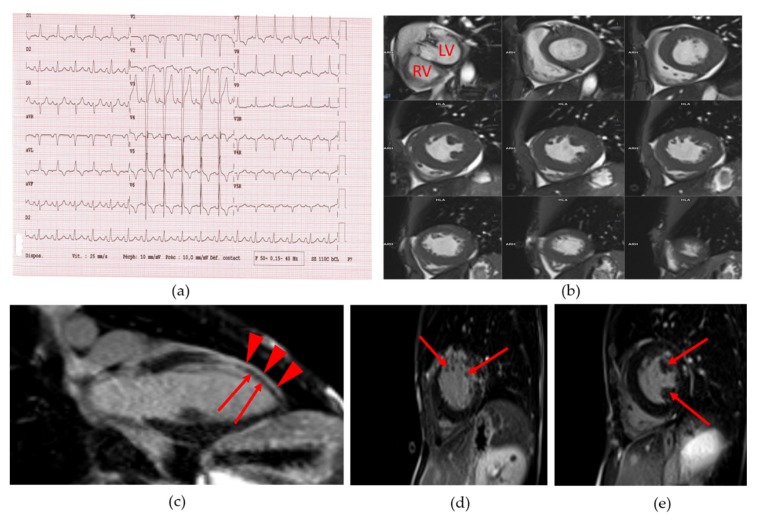
Patient 1 (**a**) 18-lead electrocardiogram (ECG) showed normal sinus rhythm with normal PR and QT intervals as well as incomplete left bundle branch bloc, left and right atrial hypertrophy and left ventricular hypertrophy; (**b**) Cine short axis cardiac magnetic resonance imaging (MRI) images, showing the left ventricular hypertrophy. LV: left ventricle. RV: right ventricle; (**c**) Cine long axis cardiac MRI image. Red arrowheads show normal gadolinium enhancement of left ventricular wall. Red arrows show defect of gadolinium staining of the antero-septo-apical myocardial region; (**d**) and (**e**): Cine short axis cardiac MRI image. Arrows show a heterogeneous late gadolinium enhancement compatible with myocardial fibrosis.

**Figure 3 cells-09-00765-f003:**
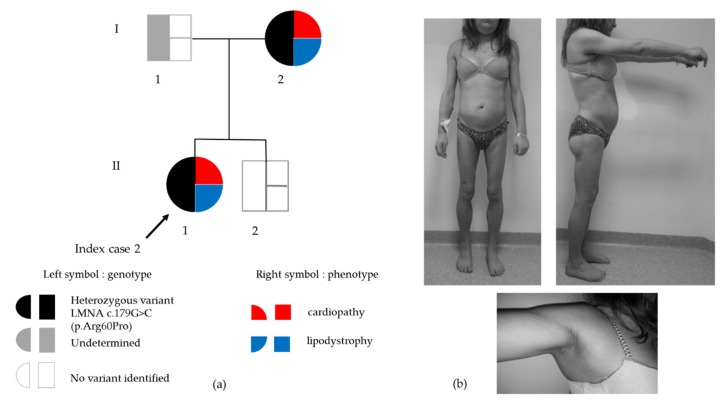
(**a**) Genealogic tree of Family 2. Black arrow indicates the index case. In the symbol for each subject, genotype is depicted on the left (black: presence of the heterozygous *LMNA* p.(Arg60Pro) variant, white: absence of the *LMNA* variant, grey: genotype not determined) and phenotype is depicted on the right (red: cardiopathy, blue: lipodystrophy); (**b**) Clinical phenotype of partial lipodystrophy in Patient 2. Upper panel: Loss of subcutaneous fat from the four limbs in contrast with accumulation of cervicofacial adipose tissue leading to a cushingoid aspect. Lower panel: axillary *acanthosis nigricans.*

**Figure 4 cells-09-00765-f004:**
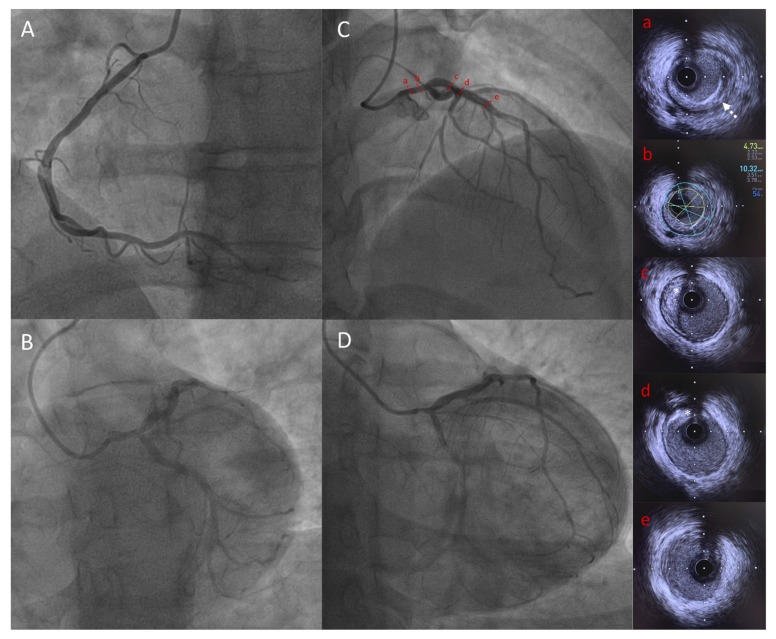
Coronary angiography and IVUS showing significant LMCA stenosis in Patient 2. Capital letters show coronary angiography: (**A**) Left anterior oblique view of right coronary artery; (**B**) Spider view of left coronary artery; (**C**) Right anterior oblique cranial view of left coronary artery showing tubular LMCA stenosis; (**D**) Right anterior oblique caudal view of left coronary artery showing tubular LMCA stenosis; Lowercase letters showing IVUS frames of LAD and LMCA: (**a**) LMCA stenosis with circumferential calcifications (white dotted arrow); (**b**) LMCA minimal lumen area measurement (4.73 mm^2^) demonstrating significant stenosis; (**c**) and (**d**) proximal LAD with calcified plaque (white star); (**e**) mid LAD showing minimal atheroma. IVUS = intravascular ultrasound, LMCA = left main coronary artery, LAD = left anterior descending artery.

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
