# Peer review of "Looking at New Unexpected Disease Targets in *LMNA*-Linked Lipodystrophies in the Light of Complex Cardiovascular Phenotypes: Implications for Clinical Practice"

_cells, 2020, doi:10.3390/cells9030765_

Round 1

Reviewer 1 Report

In this manuscript, Mosbah et al. report two cases of laminopathies caused by LMNA variants. The authors found that both patients with lipodystrophy had cardiovascular phenotypes and concluded that patients with LMNA-associated lipodystrophy should be investigated with cardiovascular clinical care.

Overall, this work is of importance to the laminopathy field because the complex multi-tissular pathophysiology is one of the traits of laminopathies. The authors also identified a novel variant of LMNA that can induce lipodystrophy and atypical progeroid syndrome which had not been reported before in case 1. However, the overlapping syndrome of lipodystrophy and cardiomyopathy has been well established in previous reports. And the sample size with 2 patients is small comparing to other case reports, especially the second patient only has mild cardiovascular phenotypes. The authors cannot exclude the possibility that the cardio dysfunction could also be caused by insulin resistant diabetes. I would suggest submit this paper to a more suitable clinical case report journal.

Author Response

We agree with the reviewer that laminopathies have a complex multi-tissular pathophysiology, and think that this is probably not sufficiently taken into account in clinical practice. Through the description and discussion of clinical cases, our paper highlights the fact that a multidisciplinary approach is particularly important for the management of laminopathies, and that cardiologists and endocrinologists should work together for each affected case.

We think that the cardiovascular phenotype of patient 2 is severe. She displayed a proximal stenosis of both the right coronary artery and the left main coronary artery leading to a severely decreased lumen area, with calcifications of the proximal left anterior descending artery, which required a triple coronary artery bypass at the age of 29. We agree that insulin resistant diabetes could contribute to the cardiac dysfunction, as discussed in the paper, but her young age, the short duration of diabetes, adequate glycemic control, normal blood pressure and LDL-cholesterol level, and the absence of tobacco use, strongly support that traditional cardiometabolic risk factors are not the only determinants of such a severe and early atherosclerotic disease.

Reviewer 2 Report

The manuscript I've been asked to review describes two LMNA-related lipodystrophic cases due to different LMNA mutations. 

As both the cases present a severe cardiovascular system involvement, and since many relatives of the patients here described died or suffered of heart diseases, to the aim of prevent harmful cardiovascular defects in other lipodystrophic patients, the authors suggest that it might be advantageous, in medical pratice to subject the lipodystrophic patients to a careful analysis of heart conditions, including a 24-hour ECG monitoring, echocardiography and/or cardiac MRI for the diagnosis of cardiomyopathy, and stress test and/or coronary CT angiogram to detect coronary artery disease. 

Besides these two cases, other lipodystrophic-related LMNA mutations causing cardiovascular system defects are already described in scientific litererature. Moreover, other laminopathies, including cardio-muscular dystrophies and progeric forms, feature severe heart defects. As a whole, the conclusions of the Authors are in my opinion, reasonable and plausible, being well supported by the data here presented and already published.   

Since the patients are described only from a clinical point of view, and since it has been demonstrated that in primary fibroblasts from lipodystophic patient presenting another LMNA mutation (p.R482L) there are evidence of an alteration of the subcellular distribution of lamin A and prelamin A (Capanni et al 2003 and Capanni et al 2005), it would be interesting in a near future to investigate the subcellular localization of mutated lamin A/C and prelamin A also in these two patients. 

Author Response

We thank the reviewer for her/his positive comments and agree that the study of the subcellular localization of lamins would be of interest in patients’ cells. We did not perform skin biopsies in these patients but will explore this possibility in order to obtain fibroblast cultures.

Reviewer 3 Report

Comments on:

 Looking at new unexpected disease targets in LMNA- linked lipodystrophies in the light of complex cardiovascular phenotypes…

This is an interesting report of 2 unusual clinical cases of young adult patients with novel LMNA mutations. This expands our understanding of the clinical consequences of LMNA mutations, which is a fairly common genetic syndrome (or perhaps more properly a  family of genetics syndromes). The conventional wisdom is that different mutations have relatively tissue-specific consequences, and this report challenges that conceptual model.

A strength of the manuscript is the detailed description of the patient pathophysiology, with a focus on cardiovascular complications. It is well written and the figures are well organized.

I don’t know about privacy laws in France but I think you need to state that these patients gave written consent to publish their de-indentified information.

It would be great to show one or two ECGs and/or Holter strips of the abnormalities.

Minor:

Line 207 “with a normal ventricular rhythm” I think I know what you mean, but maybe better to say normal PR, QRS, and QT and intervals?

Line 211 “It was electrically positive” meaning ST depressions? Please be more specific.

Line 217 “The patient underwent a hopefully successful triple coronary artery bypass graft surgery.” Delete “hopefully,” I suggest “with no acute complications”

Author Response

We thank the reviewer for her/his constructive comments.

The patients gave their written consent to publish their de-identified information, including photographs, and we have added a sentence regarding ethics as suggested.

We now show the 18-lead electrocardiogram (EKG) of patient 1, showing incomplete left bundle branch bloc, left and right atrial hypertrophy and left ventricular hypertrophy (new Fig 2a).

We have corrected all the minor points, rightly raised by the reviewer :

  • Current line 217: “PR, QRS and QT intervals were normal”
  • Lines 221-224: “The ECG stress test was stopped, as the patient developed breathlessness with a peak heart rate of 155 bpm (81% of the theorical maximum heart frequency), at 5.8 METs of work. In addition, down-sloping ST-segment depressions were observed in II, III, aVF and V3-V6 leads.”
  • Lines 229-230 “The patient underwent a successful triple coronary artery bypass graft surgery.”

Round 2

Reviewer 1 Report

I agree that in addition to the discovery a novel mutation of LMNA gene associated with laminopathy, this paper “highlights the fact that a multidisciplinary approach is particularly important for the management of laminopathies, and that cardiologists and endocrinologists should work together for each affected case”.

This paper can be considered to publish on cells.

the introduction part need to be improved by providing more known mutations of LMNA that cause cardiomyopathy and which exon of LMNA gene are most mutated in these patients. And several mutations adjacent the newly identified Asp136 site have been reported before. This need to be discussed too.